# Wasserstein diffusion on graphs with missing attributes

## Abstract

Many real-world graphs are attributed graphs where nodes are associated with non-topological features. While attributes can be missing anywhere in an attributed graph, most of existing node representation learning approaches do not consider such incomplete information. In this paper, we propose a general non-parametric framework to mitigate this problem. Starting from a decomposition of the attribute matrix, we transform node features into discrete distributions in a lower-dimensional space equipped with the Wasserstein metric. On this Wasserstein space, we propose Wasserstein graph diffusion to smooth the distribution representations of nodes with information from their local neighborhoods. This allows us to reduce the distortion caused by missing attributes and obtain integrated representations expressing information of both topology structure and attributes. We then pull the nodes back to the original space and produce corresponding point representations to facilitate various downstream tasks. To show the power of our representation method, we designed two algorithms based on it for node classification (with missing attributes) and matrix completion respectively, and demonstrate their effectiveness in experiments.

## 1 Introduction

Many real-world networks are attributed networks, where nodes are not only connected with other nodes, but also associated with features, e.g., social network users with profiles or keywords showing interests, Internet Web pages with content information, etc. Learning node representations underlie various downstream graph-based learning tasks and have attracted much attention (Perozzi et al., 2014; Grover & Leskovec, 2016; Pimentel et al., 2017; Duarte et al., 2019). A high-quality node representation is able to express node-attributed and graph-structured information and can better capture meaningful latent information.

Random walk based graph embedding approaches (Perozzi et al., 2014; Grover & Leskovec, 2016) exploit graph structure information to preserve pre-specified node similarities in the embedding space and have proven successful in various applications based on plain graphs. In addition, graph neural networks, many of which base on the message passing schema (Gilmer et al., 2017), aggregate information from neighborhoods and allow us to incorporate attribute and structure information effectively. However, most of the methods, which embed nodes into a lower-dimensional Euclidean space, suffer from common limitations: they fail to model complex patterns or capture complicated latent information stemming from the limited representation capacity of the embedding space. There has recently been a tendency to embed nodes into a more complex target space with an attempt to increase the ability to express composite information. A prominent example is Wasserstein embedding that represents nodes as probability distributions (Bojchevski & Günnemann, 2018; Muzellec & Cuturi, 2018; Frogner et al., 2019) equipped with Wasserstein metric. A common practice is to learn a mapping from original space to Wasserstein space by minimizing distortion while the objective functions are usually difficult to optimize and require expensive computations.

On the other hand, most representation learning methods highly depend on the completeness of observed node attributes which are usually partially absent and even entirely inaccessible in real-life graph data. For instance, in the case of social networks like Facebook and Twitter, in

which personal information is incomplete as users are not willing to provide their information for privacy concerns. Consequently, presentation learning models that require fully observed attributes may not be able to cope with these types of real-world networks.

In this paper, we propose a novel non-parametric framework to mitigate this problem. Starting from a decomposition of the attribute matrix, we transform node features into discrete distributions in a lower-dimensional space equipped with the Wasserstein metric and implicitly implement dimension reduction which greatly reduces computational complexity.

Preserving node similarity is a common precondition for incorporating structural information into representation learning. Based on this, we develop a Wasserstein graph diffusion process to effectively propagate a node distribution to its neighborhood and contain node similarity in the Wasserstein space. To some extent, this diffusion operation implicitly compensates for the loss of information by aggregating information from neighbors. Therefore, we reduce the distortion caused by missing attributes and obtain integrated node representations representing node attributes and graph structure.

In addition to produce distribution representations, our framework can leverage the inverse mapping to transform the node distributions back to node features (point representations). Experimentally, we show that these node features are efficient node representations and well-suited to various downstream learning tasks. More precisely, to comprehensively investigate the representation ability, we examine our framework on node classification concerning two missing cases: partially missing and entire node attributes missing. Moreover, we adapt our framework for matrix completion to show our ability to recover absent values.

**Contributions.** We develop a novel non-parametric framework for node representation learning to utilize incomplete node-attributed information. The contributions of our framework are: 1. embedding nodes into a low-dimension discrete Wasserstein space through matrix decomposition; 2. reducing distortion caused by incomplete information and producing effective distribution representations for expressing both attributes and structure information through the Wasserstein graph diffusion process; 3. reconstructing node features which can be used for various downstream tasks as well as for matrix completion.

## 2 BACKGROUND AND RELATED WORK

**Graph representation learning**

In this paper, we focus on learning node representations on attributed graphs. There are many effective graph embedding approaches, such as DeepWalk (Bojchevski & Günnemann, 2018), node2vec (Grover & Leskovec, 2016), GenVetor (Duarte et al., 2019), which embed nodes into a lower-dimension Euclidean space and preserve graph structure while most of them disregard node informative attributes. So far, there are little attention paid to attribute information (Yang et al., 2015; Gao & Huang, 2018; Hong et al., 2019). The advent of graph neural networks (Bruna et al., 2014; Kipf & Welling, 2017; Hamilton et al., 2017; Veličković et al., 2017; Gilmer et al., 2017; Klicpera et al., 2019a;b) fill the gap to some extent, by defining graph convolutional operations in spectral domain or aggregating neighborhood information in spatial domain. Their learned node representation integrate both node attributes and graph structure information.

Due to the expressive limitation of Euclidean space, embedding into a more complex target space has been explored. There are several works to leverage distributions equipped with Wasserstein distance to model complex data since a probability is well-suited for modeling uncertainty and flexibility of complex networks. For instance, Graph2Gauss (Bojchevski & Günnemann, 2018) represents node as Gaussian distributions such that uncertainty and complex interactions across nodes are reflected in the embeddings. Similarly, Muzellec & Cuturi (2018) presented a framework in which point embedding can be thought of as a particular case of elliptical distributions embedding. Frogner et al. (2019) learn to compute minimum-distortion embeddings in discrete distribution space such that the underlying distances of input space are preserved approximately.

**Missing attributes**

The completeness and adequacy of attribute information is a precondition for learning high-quality node embeddings of an attributed graph. To represent incomplete attributes in the embeddings, a fundamental method is providing plausible imputations for missing values. There is a variety of missing value imputation (MVI) techniques such as mean imputation, KNN imputation (Troyanskaya et al., 2001), softimpute (Hastie et al., 2015), and so on.

As a consequence, the representation capacity of the generated embeddings is inherently bounded by the reconstruction ability of the imputation methods. As the number of missing attributes increases, this can lead to distortion problems as well as unstable learning. The herein proposed method is the first work to compute embeddings of a graph with incomplete attributes directly.

## 3 WASSERSTEIN GRAPH DIFFUSION (WGD) FRAMEWORK

In this paper, we propose a non-parametric Wasserstein graph diffusion (**WGD**) framework on graph with missing attributes to generate node representations. The WGD framework (depicted in Fig 1) consists of three main components: 1.space transformation - to embed nodes into a Wasserstein space; 2. Wasserstein graph diffusion - to update node distribution representations; 3. inverse transformation - to pull nodes back to the original feature space.

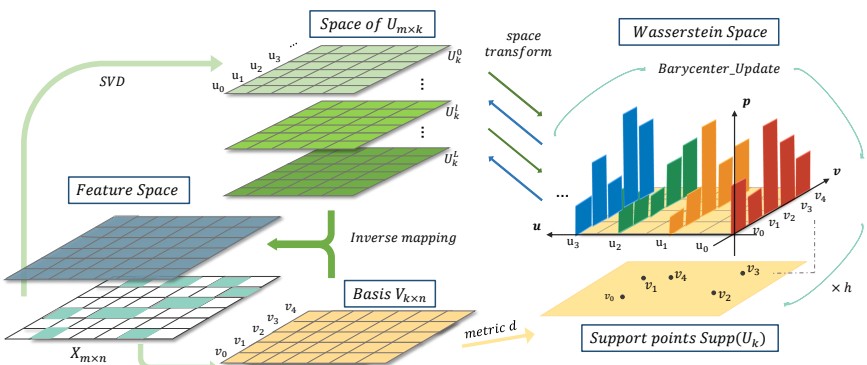

Figure 1: The WGD Framework involves transformations among three space: feature space, principal component space and discrete Wasserstein space. SVD is leveraged to generate initial node representations $U_k^0$ in principal component space which will be transformed to discrete distributions endowed with Wasserstein distance through a particular reversible positive mapping. In this discrete Wasserstein space, we utilize Wasserstein barycenter to aggregate distributional information of $h - hop$ neighbors. In each WGD layer, the updated node distributions will be transformed back to the principal component space to update corresponding node representations which is also the input of next layer. The support points are shared over layers. After $L$ times update, we pull $U_k^L$ back to the original feature space through inverse mapping to generate new node representations.

### 3.1 PRELIMINARY: WASSERSTEIN DISTANCE

Wasserstein distance is an optimal transport metric, which measures the distance traveled in transporting the mass in one distribution to match another. The **p-Wasserstein distance** between two distributions $\mu$ and $\nu$ over a metric space $\mathcal{X}$ is defined as

$$W_p(\mu, \nu) = \left( \inf_{(x,y) \sim \Pi(\mu,\nu)} \int_{\mathcal{X} \times \mathcal{X}} d(x,y)^p d\pi(x,y) \right)^{1/p}$$

where $\Pi(\mu, \nu)$ is the the set of probabilistic couplings $\pi$ on $(\mu, \nu)$, $d(x, y)$ is a ground metric on $\mathcal{X}$. In this paper, we take $p = 2$. The **Wasserstein space** is a metric space that endow probability distributions with the Wasserstein distance.

### 3.2 The space transformation

Space transformation is the first step of our WGD framework, which attempts to transform node features to discrete distributions endowed with the Wasserstein metric.

A common assumption for matrix completion is that the matrix is low-ranked, i.e. the features lie in a smaller subspace, and the missing features can be recovered from this space. Inspired by Alternating Least Square(ALS) Algorithm, a well known missing value imputation method which follows this assumption and uses SVD to factorize matrix into low-ranked submatrices, we first decompose the feature matrix $X \in R^{n \times m}$ into a principal component matrix $U$, an singular value matrix $\Lambda$, and an orthogonal basis matrix $V$, i.e. $X = U\Lambda V^{\top}$. For dimensionality reduction, we only account for the first $k$ singular vectors of $V$:

$$U_k, \Lambda_k, V_k = \text{SVD}(X, k), \tag{1}$$

here $U_k \in R^{n \times k}$, $V_k \in R^{m \times k}$ and $\Lambda_k \in R^{k \times k}$. To impute missing entries, ALS alternatively optimizes these submatrices. While our method is not for matrix completion and not need to optimize $U_k$ and $V_k$. We aim to generate expressive node representations in the principal components space where $U_k$ is the initial node embedding matrix. It is worth noting that such node representations have strong semantic information: each feature dimension is a basis vector from $V_k$. Moreover, in a broad sense (allowing the existence of negative frequencies), we can express nodes as general histograms with principal components (the rows of $U_k$, noted as $row(U_k)$), acting as frequencies and basis vectors (the columns of $V_k$, noted as $col(V_k)$) acting as bins. Therefore, to capture the ground semantic information, we transform $U_k$ from a Euclidean space to a discrete distribution space. More precisely, this space transformation involves a reversible positive function (here we use the exponential function for implementation) and a normalization operation:

$$\tilde{U}_k := \phi(U_k) = \text{Normalize}(\exp(\text{row}(U_k))). \tag{2}$$

Here, $\tilde{U}_k$ can be seen as the discrete distributions of nodes which have common support points $col(V_k)$ with the notation $supp(U_k)$, i.e. $supp(U_k) = col(V_k)$. Each row of $\tilde{U}_k$ is a discrete distribution with the form $u_i = \sum_j a_{ij}\delta_{v_j}$ where $a_{ij}$ are weigths summing to 1, and $v_j$ is a column of $V_k$ as well as a support point. The distance of various support points, noting that $\Lambda$ is the square root of the eigenvalues of $X^{\top}X$, i.e. we define the ground metric $d$ as follows:

$$d(v_i, v_j) = \| X(v_i - v_j) \|^2 = \| \Lambda_{ii}v_i - \Lambda_{jj}v_j \|^2 = |\Lambda_{ii}^2 - \Lambda_{jj}^2|. \tag{3}$$

Here, $v_i$ and $v_j$ refer to the $i$-th and $j$-th support points which are mutual orthogonal unit vectors. In the meantime, we equip node discrete distributions $\tilde{U}_k$ with Wasserstein metric:

$$W_2^2(\tilde{u}_i, \tilde{u}_j | D) = \min_{T \geq 0} tr(DT^{\top}) \quad \text{subject to } T\mathbf{1} = \tilde{u}_i, \ T^{\top}\mathbf{1} = \tilde{u}_j. \tag{4}$$

Here $D \in R^{k \times k}$ refers to the underlying distance matrix with $D_{ij} = d(v_i, v_j)$.

### 3.3 The Wasserstein graph diffusion process

Although we produce node distribution representations from space transformation, the information extracted from such representations is limited and distorted caused by missing attributes. To reduce the distortion, we carry out the Wasserstein graph diffusion process to smooth node distributions over its local neighborhood such that valid information extracted from different node attributes can be shared, i.e. informational complementarity. The Wasserstein graph diffusion can boil down to an aggregation operation realized by computing Wasserstein barycenter that is exactly the updated node representations. In this way, both topology structure information and the aforementioned integrated attribute information are incorporated into node representations. Note that it is similar to the message aggregation in graph neural networks, except we do it in a Wasserstein space and introduce no parameters.

We denote the node distribution update process as Barycenter Update:

$$\tilde{u} = Barycenter\_Update(u, D) := \arg\inf_{p} \frac{1}{|N(u)|} \sum_{u' \in N(u)} W_2^2(p, \tilde{u}' | D), \tag{5}$$

here $\overline{N(u)} = N(u) \cup \{u\}$, $N(u)$ refers to the neighbors of $u$, $|\overline{N(u)}|$ equals to the degree of $u$ (with self-loop) and $\tilde{u}'$ refers to the discrete distribution of node $u'$. Similar to common message aggregation on graph, the Wasserstein diffusion process included $l$ times Barycenter Update can affect $l$-hop neighbors.

Recall that the set of support points of barycenter, noted as $S_u$, contains all possible combination of the common support points $supp(U_k)$ shared by all initial node distributions:

$$S_v = \{ \frac{1}{|\overline{N(u)}|} \sum_{i=1}^{|\overline{N(u)}|} x_i | x_i \in \text{supp}(U_k)\}. \tag{6}$$

Noting that, as the Barycenter Update process goes on, nodes will not share the same support points any more and their set of support points will be larger and larger that dramatically increase the computation. On the other hand, such a free-support problem is notoriously difficult to solve. Therefore, we leverage the fixed-support Wasserstein barycenter for the update that preserves the initial support $supp(U_k)$. Therefore, no matter how many times we update node distributions, they always share the common support points. In practice, we use Iterative Bregman Projection (IBP) algorithm (Benamou et al., 2015) to obtain the fixed-support barycenter (see Algorithm 1 ). The Wasserstein diffusion process is formulated as follows:

$$\tilde{U}_k^{(0)} = \phi(U_k); \quad \tilde{U}_k^{(l+1)} = \text{Barycenter\_Update}(\tilde{U}_k^{(l)}, D) \text{ with fixed supp}(U_k). \tag{7}$$

Through the Wasserstein diffusion process, we obtain the updated node distributions $\tilde{U}_k$ which have an effective ability to represent node attributes and graph structure.

---

**Algorithm 1** Iterative Bregman Projection for Barycenter Update

---

    **input:** discrete distribution matrix $P_{d \times n}$, distance matrix $D_{d \times d'}$, weights vector $w$, $\epsilon$.
1: **initialize:** $K = exp(-D/\epsilon)$, $V_0 = \mathbf{1}_{d' \times n}$
2: **for** $i = 1 \ldots M$ **do**
3:     $U_i = \frac{P}{KV_{i-1}}$
4:     $p_i = exp(log(K^\top U_i)w)$
5:     $V_i = \frac{p_i}{K^\top U_i}$
6: **end for**
7: **output:** $p_M$

---

### 3.4 THE INVERSE TRANSFORMATION

We first derive an approximate inverse transformation from the prespecified mapping (2) to convert the updated distribution representations $\tilde{U}_k$ to the principal component space:

$$\bar{U}_k = \text{Gram\_Schmidt\_Ortho}(\text{col}(\log(\tilde{U}_k))). \tag{8}$$

Here Gram_Schmidt_Ortho refers to the Gram-Schmidt Orthogonalization processing used to hold the semantic structure of the principal component space. In addition, as a side effect, empirical results show that orthogonalizing node embeddings can efficiently alleviate over-smoothing problem.

On the other hand, SVD separates the observed attribute information into two parts: one is extracted by the updated node distributions while the other maintains in the fixed support points. To generate more expressive node representation incorporating both the two parts of information, we pull nodes back to the feature space:

$$\tilde{X} = \bar{U}_k \Lambda_k V_k^\top. \tag{9}$$

Note that this $\tilde{X}$ is not a matrix completion for the original $X$, since the elements in $X$ do not remain the same. Instead, it is a transformation of the representation $U_k$ after our Wasserstein diffusion process, by combing with neural networks it can be then used for various downstreaming tasks, as we will show in the next section.

# 4 EMPIRICAL STUDY

As explained in the previous section, our WGD framework can incorporate node attributes and graph structure to reconstruct node representations in the original space, i.e. the feature matrix. Therefore, it is well-suited for various downstream learning tasks as well as matrix completion. In this section, we adapt WGD for node classification tasks and matrix completion then evaluate the quality of reconstructed node features (representations) respectively.

## 4.1 NODE CLASSIFICATION ON GRAPHS WITH MISSING ATTRIBUTES

In this section, we apply our framework to node classification tasks on attributed graphs with missing features. Algorithm 2 summarizes the architecture. Each WGD layer (the outer loop) contains three main stages: space transformation, Wasserstein diffusion, and inverse transformation. The explicit formulas of space transformation and inverse transformation are given by (2) and (8). In the diffusion process, we conduct Barycenter_Update (7) $h$ times to aggregate $h$-hop neighbors probability information and take the mean outputs of each layer as the updated principal components matrix to avoid over smoothing. We reconstruct feature matrix (9) (acting as updated node representations) on the original space and feed them to a two-layer MLP for node classification, termed **WGD-MLP**.

To confirm the importance of the Wasserstein diffusion process, we propose an ablation framework, called **SVD-GCN** , in which we skip the diffusion process and remain the other stages unchanged. The forward formula of $l+1$-th SVD-GCN layer is:

$$X^{(l+1)} = U_k^{(l)} \Lambda_k V_k^{(l)^\top}, \quad with \ U_k^{(l)}, \ \Lambda_k, \ V_k^{(l)} = \text{SVD}(X^{(l)}, k).  \tag{10}$$

Precisely, we leverage low-rank SVD to factorize the matrix then feed the reconstructed feature matrix to a two-layer GCN (Kipf & Welling, 2017). Furthermore, to show that matrix decomposition is necessary, we provid an additional ablation framework, termed **WE-MLP**. In this baseline, we directly transform node features to discrete distributions skipping the stage of matrix decomposition. The forward formula of $l+1$-th WE-MLP layer is:

$$X^{(l+1)} = \log(\text{Barycenter\_Update}(\text{Normalize}(\exp(X^{(l)})))).  \tag{11}$$

After that, we feed the output that is identical to the updated node representations to a two-layer MLP for node classification.

---

**Algorithm 2** WGD adapted for node classification on graphs with missing attributes

> **input:** attribute matrix $X_{n \times m}$ containing initial values, $k$.
1: Apply $k$-rank SVD to $X$: $U_k, \Lambda_k, V_k = \text{SVD}(X, k)$
2: $U_k^{(0)} \leftarrow U_k$
3: **for** $l = 1 \rightarrow L$ **do**
4:     *space transformation:* $\tilde{U}_k^{(l-1)} = \text{Normalize}(\exp(\text{row}(U_k^{(l-1)})))$
5:     $\hat{U}_k^{(0)} \leftarrow \tilde{U}_k^{(l-1)}$
6:     **for** $i = 1 \rightarrow h$ **do**
7:         *Wasserstein diffusion:* $\hat{U}_k^{(i)} = \text{Barycenter\_Update}(\hat{U}_k^{(i-1)}, D)$
8:     **end for**
9:     *inverse transformation:* $U_k^{(l)} = \text{Gram\_Schmidt\_Ortho}(\text{col}(\log(\hat{U}_k^{(h)})))$
10: **end for**
11: $\bar{U}_k = mean(U_k^{(1)}, ..., U_k^{(L)})$
12: $\tilde{X} = \bar{U}_k \Lambda_k V_k^\top$
13: Apply MLP for node classification: $Y = \text{MLP}(\tilde{X})$

---

**Experimental Settings.** In the experiments of node classification, we implicitly evaluate the ability of WGD to produce high-quality node representations utilizing incomplete attribute information. Since node attributes of common benchmarks of node classification, such as citation networks, are usually fully collected, we artificially remove some attributes at random. To thoroughly and quantitatively evaluate our representation capacity, we

conducte experiments on three common node classification benchmarks: **Cora**, **Citeseer** and **PubMed** in two settings:

**a. Partially Missing** Partially missing means some entities of the feature matrix are missing. Precisely, we randomly remove the values of an attribute matrix in a given proportion (from 10% to 90%).

**b. Entirely Missing** Entirely missing means some nodes have complete attributes while the others have no feature at all. In this case, we remove some rows of the feature matrix in a given proportion (from 10% to 90%) at random.

We apply various traditional imputation approaches including zero-impute, mean-impute, soft-impute (Mazumder et al., 2010), and KNN-impute (Batista et al., 2002) to complete the missing values and use a two-layer GCN to classify nodes based on imputed feature matrices. We call them **ZERO-GCN**, **MEAN-GCN**, **SOFT-GCN** and **KNN-GCN** respectively. Here, zero-impute and mean-impute means that we replace missing values with zero value or the mean of observed features respectively. Moreover, we set the performance of GCN only leveraging graph structure, i.e. with identity matrix as feature matrix, as a lower bound, named **GCN_NOFEAT**. Moreover, in this situation, we use Label Propagation (**LP**) Algorithm as a better low-bound. If the performance of a model is below the low-bounds, it reflects that the model is unable to utilize incomplete attributes effectively.

In all the benchmark and experimental settings, we fix most of the hyperparameters of WGD-MLP: 7 WGD layers including twice Barycenter_Update on each layer, two-layer MLP with 128 hidden units with 0.5 dropout. We apply rank 64 SVD on the incomplete attribute matrix in which we replace the missing values with the mean value of observed features. The settings of GCN used in all baselines and SVD-GCN are the same, i.e. 2 layers with 32 hidden units. For all models, we use Adam optimizer with 0.01 learning rate and weight decay depending on different missing cases. We early stop the model training process with patience 100, select the best performing models based on validation set accuracy, and report the mean accuracy for 10 runs as the final results. For more experimental details please refer to our codes: https://anonymous.4open.science/r/3507bfe0-b3b1-4d18-a7b2-eb3643ceedb1

**Experimental Results** As shown in Fig 2, WGD-MLP outperforms all baselines and

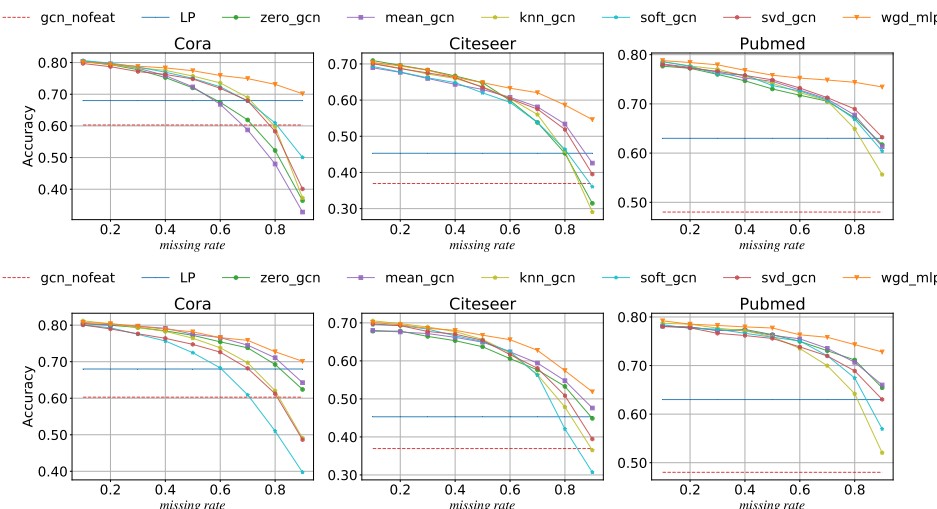

Figure 2: Average accuracy of models in entirely missing (top) and partially missing (bottom) settings.

achieves significant improvement, especially when the missing rate is over 50%. There is a clear trend of decreasing of baselines in two types of missing attributes. In contrast, the flat curve of WGD-MLP reflects our remarkable robustness. On Cora and Citeseer, comparison of the lower bound provided by GCN_NOFEAT with baselines confirms that imputation strategies are ineffective and even counterproductive when observed attributes are grossly

inadequate. By contrast, WGD always shows a continuously satisfactory performance, even when 90% of attributes are unavailable. For instance, in both two types of missing data, the performance of WGD consistently falls by only 10% and 5%, respectively, on Cora and Pubmed datasets. It illustrates that WGD can significantly reduce informative distortion caused by missing data and learn effective latent node representations. On the other hand, the inferior performance of SVD-GCN convincingly demonstrates the efficiency of Wasserstein diffusion. For WE-MLP, we conduct experiments on Cora and Pubmed with a fixed missing rate (0.1). Here is the results: Cora: 0.453 (entirely missing), 0.439 (partially missing); Pubmed: 0.575 (entirely missing), 0.537 (partially missing). It implies a clear benefit of matrix decomposition in the first step of WGD.

**Sensitive Analysis** To show how the number of WGD layers $L$ and iterations $h$ of Wasserstein diffusion in each layer influence our performance of node classification, we take Cora as an example and present the results in Fig 3. As we know, many GNN models encounter the over-smoothing issue when they go deep; however, Figure 3 shows contrary results and demonstrates that our method can efficiently handle over-smoothing. This is due to two strategies: self-connection (similar to the strategy in APPNPKlicpera et al. (2019a) ) and orthogonalization which make nodes different.

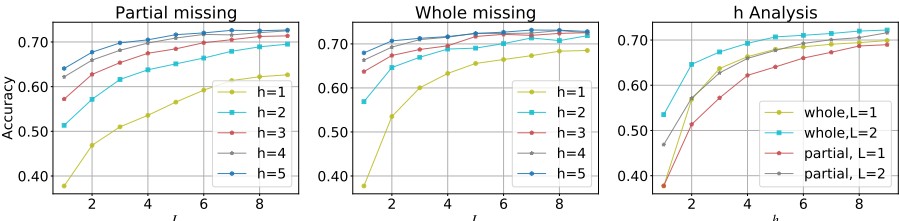

Figure 3: Sensitive analysis for the number of WGD layers $L$ and times $h$ of Wasserstein diffusion in each layer. The results (node classification accuracy on Cora dataset) show that our method prevents over-smoothing.

## 4.2 Multi-Graph Matrix completion

An interesting aspect of the WGD framework is that they allow us to reconstruct feature matrix with the introduction of reconstruction constrains and neural networks. In this section, we test the ability of WGD to reconstruct missing values of a multi-graph matrix, or more precisely, of recommendation systems with additional information including the similarity of users and items represented as the users graph and the items graph, respectively. Algorithm 3 summarizes the WGD framework adapted for matrix completion tasks.

Assume that $X$ is the incomplete feature matrix of the users graph, then $X^\top$ is that of the items graph. Leveraging $k$-rank SVD, we have $U_k, \Lambda_k, V_k = \mathrm{SVD}(X, k)$. Through the space transformation (2), we obtain $\phi(U_k)$, the discrete distributions of users with $supp(U_k) = col(V_k)$. In the same way, we have $V_k, \Lambda_k, U_k = \mathrm{SVD}(X^\top, k)$ and $\phi(V_k)$ with $supp(V_k) = col(U_k)$. Typically, the update of node distributions on one graph will lead to unexpected changes in the support points of the other graph. However, this does not appear to be the case in WGD, as the predefined distance matrix $D$ (3) only depends on the fixed $\Lambda_k$. Therefore, $supp(U_k)$ and $supp(V_k)$ share a common distance matrix. It implies that, two Wasserstein diffusion processes formulatd by (5), can simultaneously go on. Informally, this generalized WGD framework, called the **multi-graph WGD**, can be thought of as an overlay of two original WGDs. The difference appears in the last step: in the multi-graph WGD, we concatenate the outputs of each layer, feed them to a simple MLP and normalize columns of the learned $U_k$ and $V_k$ to be unit vectors.

**Benchmarks.** We conduct experiments on two popular matrix completion datasets with multi-graph: Flixster and MovieLens-100K processed by Monti et al. (2017).

**Baselines.** We compare our multi-graph WGD framework with some advanced matrix completion methods, including **GRALS** (Rao et al., 2015), **sRMGCNN** (Monti et al., 2017), **GC-MC** (Berg et al., 2017), **F-EAE** (Hartford et al., 2018), and **IGMC** (Zhang

---

**Algorithm 3** multi-graph WGD adapted for matrix completion

**input:** attribute matrix $X_{n \times m}$ containing initial values, $k$.

1: Apply $k$-rank SVD to $X$: $U_k, \Lambda_k, V_k = \text{SVD}(X, k)$
2: $U_k^{(0)} \leftarrow U_k, V_k^{(0)} \leftarrow V_k$
3: **for** $l = 1 \rightarrow L$ **do**
4:     *space transormation:* $\tilde{U}_k^{(l-1)} = \text{Normalize}(\exp(\text{row}(U_k^{(l-1)})))$
5:     *space transormation:* $\tilde{V}_k^{(l-1)} = \text{Normalize}(\exp(\text{row}(V_k^{(l-1)})))$
6:     $\hat{U}_k^{(0)} \leftarrow \tilde{U}_k^{(l-1)}, \hat{V}_k^{(0)} \leftarrow \tilde{V}_k^{(l-1)}$
7:     **for** $i = 1 \rightarrow h$ **do**
8:         *Wasserstein diffusion:* $\hat{U}_k^{(i)} = \text{Barycenter\_Update}(\hat{U}_k^{(i-1)}, D)$
9:         *Wasserstein diffusion:* $\hat{V}_k^{(i)} = \text{Barycenter\_Update}(\hat{V}_k^{(i-1)}, D)$
10:     **end for**
11:     *inverse transformation:* $U_k^{(l)} = \text{Gram\_Schmidt\_Ortho}(\text{col}(\log(\hat{U}_k^{(h)})))$
12:     *inverse transformation:* $V_k^{(l)} = \text{Gram\_Schmidt\_Ortho}(\text{col}(\log(\tilde{V}_k^{(h)})))$
13: **end for**
14: $\hat{U}_k = concat(U_k^{(0)}, ... U_k^{(L)}), \hat{V}_k = concat(V_k^{(0)}, ... V_k^{(L)})$
15: *Reconstruct $U_k$:* $\bar{U}_k = L_2\_normalize(MLP_u(\hat{U}_k))$
16: *Reconstruct $V_k$:* $\bar{V}_k = L_2\_normalize(MLP_v(\hat{V}_k))$
17: **return** $\bar{X} = \bar{U}_k \Lambda_k \bar{V}_k^\top$

---

Table 1: RMSE test results on Flixster and MovieLens-100K.

|  | GRALS | sRMGCNN | GC-MC | F-EAE | IGMC | WGD-MLP |
|---|---|---|---|---|---|---|
| FLIXSTER | 1.313 | 1.179 | 0.941 | 0.908 | 0.872 | 0.883 |
| ML-100K | 0.945 | 0.929 | 0.910 | 0.920 | 0.905 | 0.910 |

& Chen, 2020). GRALS is a graph regularization method and sRMGCNN is a factorized matrix model. GC-MC directly applies GCN for link prediction on the user-item bipartite graph. F-EAE and IGMC are inductive matrix completion methods without using side information. The former leverages exchangable matrix layers while the latter focus on local subgraphs around each rating and trains a GNN to map the subgraphs to ratings.

**Experimental Settings and Results.** We follow the experimental setup of Monti et al. (2017) and take the common metric Root Mean Square Error (RMSE) to evaluate the accuracy of matrix completion. We set rank= 10 as the same as sRMGCNN for Flixster and MovieLens-100K. In addition, we use 4-layer MLP with 160 hidden units in all experiments. We set $L = 5$, $h = 1$ in Flixster and $L = 7$, $h = 1$ in MovieLens-100K. We train the model using Adam optimizer with 0.001 learning rate. Table 1 presents the experimental results. As we can see, our WGD-MLP model outperforms most methods and achieves comparable performance as the state of art model IGMC. However, the parameter numbers in our model is much less than the GNN based model IGMC.

## 5 CONCLUSION

Graphs with missing node attributes are ubiquitous in the real-world, while most node representation learning approaches have limited ability to capture such incomplete information. To mitigate this problem, we introduced matrix decomposition and Wasserstein graph diffusion for representation learning such that observed node features can be transformed into discrete distributions and diffused along the graph. We developed a general framework that can produce high-quality node representations with powerful ability to represent attribute and structure information and adapted the framework for two applications: node classification and matrix completion. Extensive experiments on node classification under two missing settings verified the powerful representation capacity and superior robustness of our framework. Our model also proves effective to recover missing features in matrix completion.

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
